# Conical Statistical Optimal Near-Field Acoustic Holography with Combined Regularization

**DOI:** 10.3390/s21217150

**Published:** 2021-10-28

**Authors:** Wei Cheng, Jinglei Ni, Chao Song, Muhammad Mubashir Ahsan, Xuefeng Chen, Zelin Nie, Yilong Liu

**Affiliations:** 1State Key Laboratory for Manufacturing Systems Engineering, Xi’an Jiaotong University, Xi’an 710049, China; nijinglei@stu.xjtu.edu.cn (J.N.); songch@stu.xjtu.edu.cn (C.S.); mubashirahsan@stu.xjtu.edu.cn (M.M.A.); chenxf@xjtu.edu.cn (X.C.); zlnie@sei.xjtu.edu.cn (Z.N.); yilong@xjtu.edu.cn (Y.L.); 2School of Mechanical Engineering, Xi’an Jiaotong University, Xi’an 710049, China

**Keywords:** sound field reconstruction, conical statistical optimal near-field acoustic holography (SONAH), truncated singular value decomposition, combined regularization method, noise monitoring and control

## Abstract

For the sound field reconstruction of large conical surfaces, current statistical optimal near-field acoustic holography (SONAH) methods have relatively poor applicability and low accuracy. To overcome this problem, conical SONAH based on cylindrical SONAH is proposed in this paper. Firstly, elementary cylindrical waves are transformed into those suitable for the radiated sound field of the conical surface through cylinder-cone coordinates transformation, which forms the matrix of characteristic elementary waves in the conical spatial domain. Secondly, the sound pressure is expressed as the superposition of those characteristic elementary waves, and the superposition coefficients are solved according to the principle of superposition of wave field. Finally, the reconstructed conical pressure is expressed as a linear superposition of the holographic conical pressure. Furthermore, to overcome ill-posed problems, a regularization method combining truncated singular value decomposition (TSVD) and Tikhonov regularization is proposed. Large singular values before the truncation point of TSVD are not processed and remaining small singular values representing high-frequency noise are modified by Tikhonov regularization. Numerical and experimental case studies are carried out to validate the effectiveness of the proposed conical SONAH and the combined regularization method, which can provide reliable evidence for noise monitoring and control of mechanical systems.

## 1. Introduction

Vibration and noise have a great influence on the accuracy of mechanical processing and reliability of electromechanical products, especially stealth and detection performance of underwater vehicles. The primary problem in the evaluation of stealth performance of underwater vehicles is the acquisition of radiated sound field. However, it is difficult to carry out high-precision tests in the far fields. Therefore, near-field acoustic holography (NAH) is applied to gain three-dimensional (3D) visualization of sound radiation based on measurements over a surface near the sound source, whose ability to reconstruct the evanescent wave components also ensures a very high spatial resolution.

The traditional NAH [1] is based on regular-grid measurements across a level surface in a separable coordinate system, allowing the calculation to be performed by spatial discrete Fourier transform (DFT). Consequently, the main limitation is the requirement for full coverage of the areas with significant sound pressure. A second limitation is the requirement of a regular measurement grid to support the use of spatial DFT. Both above limitations are overcome by the Patch NAH technique named Statistical Optimal Near-Field Acoustic Holography (SONAH), which provides a new way of better sound-field reconstructions directly by setting up a projection matrix that is optimized for a wave-number spectrum of wave functions with a specific amplitude distribution (scaling), avoiding the truncation effects and winding error caused by DFT of the traditional NAH. Nevertheless, most current methods focus on the statistical optimal planar near-field acoustic holography and only a few studies on the cylindrical sound source for military equipment have been carried out [2,3,4]. When the submarine travels at a low speed, its radiation noise mainly comes from the mechanical noise of the conical stern. Currently, there is a lack of research on the reconstruction of the sound field radiated by conical sources and the lack of research on acoustic inverse problems still exists. Therefore, the conical SONAH demands immediate attention, so that it can be applied to the conical shell underwater weapons. On this basis, based on cylindrical SONAH, cylindrical SONAH with the combined regularization method is proposed to solve the inadequacy of inverse problems and strengthen the stability and accuracy of the reconstruction process.

Near-field acoustic holography (NAH) developed in the 1980s is a very useful tool for three-dimensional (3D) visualization of sound radiation and for precise source localization. The traditional NAH [5,6] is based on regular-grid measurements across a surface in a separable coordinate system, allowing the calculations to be performed by spatial discrete Fourier transform (DFT), causing the spatial truncation effects and winding error. Therefore, a set of techniques have been developed to avoid the spatial truncation effects. One of them is Helmholtz Equation least-squares (HELS) [7,8] and another similar method is the Equivalent Source Method (ESM) [9,10,11,12]. A third much-related technique is the patch inverse boundary element method (BEM) [13,14,15]. The fourth and last technique that should be mentioned here is statistically optimized NAH (SONAH) [16,17,18], which can reconstruct the sound field in a partial holographic surface and overcome strict requirements for the size of the measurement surface. Jacobsen [19] confirmed that SONAH with advantage can be based on measurement of the sound pressure and the normal component of the particle velocity to overcome the limitation, that all sources should be on one side of the measurement plane whereas the other side must be source free. Hald [20] provided an overview of the basic theory of SONAH and investigated the sensitivity of the inherent error distribution with changes in the parameters of the SONAH method. Kim [21] proposed an improved SONAH procedure and reconstructed the sound locations and radiation patterns of the two loudspeakers in a subconical moving fluid medium. Zhu [22] analyzed a modulated sound source with SONAH technology and applied it to loudspeaker tests and air compressor tests. Hald [23] introduced scaling of the applied plane wave functions that took the evanescent wave amplification into account to support the regularization methods in finding the best compromise between noise suppression and reconstruction accuracy. However, most of the existing research has focused on the statistically optimized planar NAH (SOPNAH). Cho [24] visualized source regions of a refrigeration compressor accurately by using fewer measurement positions than conventional NAH. Yang [25] identified the noise sources in underwater cylinder based on measurement of particle velocity in the simulation and experiments. Wall [26] used the inclusion of multiple wave-function to modify SONAH and obtained an accurate near-field reconstruction. Later, Wall [27] combined an equivalent wave model (EWM) and SONAH for the reconstruction of near-field pressures in multisource environments with lower errors and fewer measurements, which was used to reconstruct apparent source distributions between 20 and 1250 Hz at four engine powers and produced accurate field reconstructions for both inward and outward propagation [3]. Nevertheless, most of these existing studies focused on planar sound sources, and only some focused on cylindrical sound sources. Furthermore, currently, there is a lack of research on the reconstruction of sound fields radiated by conical sources aiming at high accuracy that are suitable for conical shells of underwater vehicles.

NAH is a linear, ill-posed inverse problem due to the existence of strongly decaying evanescent waves. Regularization provides a method of generating a solution to the linear problem in an automated way. Therefore, regularization methods are often used to overcome ill-posed problems in NAH and the regularization parameter selection is very important. Saijyou [28] proposed an estimation method of appropriate regularization filter for applying the K-space data extrapolation method to solve the problem of the measured pressure contaminated by noise. Pascal [29] determined the regularization parameter using the Morozov discrepancy principle in SONAH and made significant improvements to the standard NAH. Gomes [30] compared the three regularization parameter choice methods (PCMs) used in NAH: GCV, L-curve and Normalized Cumulative Periodogram (NCP). He [31] first combined the two most commonly used regularization methods, Tikhonov regularization and truncated singular value decomposition (TSVD), into ESM-based NAH. However, all these regularizations were investigated only in NAH and SOPNAH and it is difficult to obtain a stable and meaningful solution for conical SONAH by a single regularization method. Therefore, it is necessary to study a new regularization method for applying conical SONAH to strength the stability of reconstruction.

In the existing research, our group developed some related researches which provided some important evidence for noise monitoring and control [32,33]. On this basis, conical SONAH is proposed in this paper to realize the identification and localization of noise source to gain three-dimensional (3D) visualization of sound radiation of the conical sound source. In this method, the elementary cylindrical wave is transformed into the characteristic elementary wave suitable for the radiated sound field of a conical surface through the cylinder-cone orthogonal coordinate transformation method, forming the matrix of characteristic elementary wave in the conical spatial domain. Then, the sound pressure is expressed as the superposition of this characteristic elementary wave. Next, the superposition coefficient is solved according to the principle of superposition of wave field. Lastly, the reconstructed conical pressure is expressed as a linear superposition of the holographic conical pressure. Besides, aiming at the above ill-posed inverse problem, a combined regularization method that combines the advantages of TSVD and Tikhonov regularization is proposed to further improve reconstruction accuracy, where the different singular values of the acoustic transfer matrix are dealt differently. The low-frequency component without noise corresponding to the larger singular values is processed by the TSVD method, and the smaller singular values corresponding to the high spatial frequency component containing noise in the measurement are processed by Tikhonov regularization. The stability and accuracy of the proposed method were verified through a set of computer simulations. Subsequently, an experiment was designed in the anechoic chamber to validate the effectiveness of the proposed method and combined regularization method.

The remainder of the paper is organized as follows: In Section 2, the basic implementation procedure of conical SONAH with combined regularization method is provided. In Section 3, numerical case studies on conical SONAH are provided to study the influence of cone angle on the accuracy of acoustic field reconstruction and the performances are comparatively studied with the conventional cylindrical SONAH. The combined regularization method is also comparatively studied with the traditional single regularization method. In Section 4, a test bed with shell structures and the test implementation steps are introduced, and the performances of the proposed conical SONAH and combined regularization method are also given. In Section 5, the conclusions are summarized. Generally, this study is intended to provide a reference for the engineering application of conical SONAH and can benefit vibration and noise monitoring, reduction, and control for mechanical systems.

## 2. Theories of Conical SONAH

### 2.1. Implementation Process of Conical SONAH

The conical SONAH is based on the cylindrical SONAH. First, the elementary cylindrical wave is transformed into the characteristic elementary wave suitable for the radiated sound field of a conical surface through the cylinder-cone orthogonal coordinate transformation method, and the matrix of characteristic elementary wave in the conical spatial domain is formed. Then, the sound pressure is expressed as the superposition of this characteristic elementary wave. Next, the superposition coefficient is solved according to the principle of superposition of the wave field with the combined regularization method. Lastly, the reconstructed conical pressure is expressed as a linear superposition of the holographic conical pressure. The implementation process of conical SONAH is given in Figure 1. 

The detailed implementation process of conical SONAH is as follows:

From the Equation of acoustic waves of small amplitude in an ideal fluid medium, the Helmholtz Equation of a time-independent and single-frequency acoustic field can be obtained [24].
(1)$$\nabla^{2}p(x,y,z)+k^{2}p(x,y,z)=0$$
where p(x,y,z) is the spatial sound pressure and it is a function of the spatial position, k=ω/c=2π/λ is the number of sound waves, c is the sound speed, λ is the wavelength.

For Equation (1), cartesian coordinates can be converted into a cylindrical coordinate system by ordering x=rcosθ,y=rsinθ. The Laplace operator can be expressed as a cylindrical coordinate:(2)$$\nabla^{2}=\frac{\partial^{2}}{\partial r^{2}}+\frac{1}{r}\frac{\partial }{\partial r}+\frac{1}{r^{2}}\frac{\partial^{2}}{\partial \theta^{2}}+\frac{\partial^{2}}{\partial z^{2}}$$

In this cylindrical coordinate, Equation (1) is solved by the separation variable method, and the form of the solution is:(3)$$p(r,\theta ,z)=p_{r}(r)p_{\theta }(\theta )p_{z}(z)$$
where p(r,θ,z) is the spatial sound pressure in cylindrical coordinate system. By substituting Equation (2) into the Helmholtz Equation in the cylindrical coordinate, it becomes:(4)$$\frac{1}{p_{r}}\frac{d^{2}p_{r}}{d^{2}r}+\frac{1}{rp_{r}}\frac{dp_{r}}{dr}+\frac{1}{r^{2}p_{\theta }}\frac{d^{2}p_{\theta }}{d\theta^{2}}+\frac{1}{p_{z}}\frac{d^{2}p_{z}}{dz^{2}}+k^{2}=0$$

Because $p_{z}$ is only related with coordinates z in the above Equation, so it can be set to be equal to the constant $-k_{z}^{2}$, the result is shown below:(5)$$\frac{1}{p_{z}}\frac{d^{2}p_{z}}{dz^{2}}+k_{z}^{2}=0$$Similarly, the item only related with angle can *θ* be expressed as:(6)$$\frac{1}{p_{\theta }}\frac{d^{2}p_{\theta }}{d\theta^{2}}+n^{2}=0$$

Substitute Equations (5) and (6) into the Equation (4) to obtain:(7)$$\frac{d^{2}p_{r}}{dr^{2}}+\frac{1}{r}\frac{dp_{r}}{dr}+(k_{r}^{2}-\frac{n^{2}}{r^{2}})p_{r}=0$$

In the Equation, when $k_{z}^{2}\leq k^{2}$,
(8)$$k_{r}=\sqrt{k^{2}-k_{z}^{2}}$$
when $k_{z}^{2}>k^{2}$,
(9)$$k_{r}=i\sqrt{k_{z}^{2}-k^{2}}$$

The traveling wave solution of the Helmholtz Equation in the cylindrical coordinate system can be obtained as:(10)$$p(r,\theta ,z)=\sum_{n=-\infty }^{+\infty }e^{in\theta }\frac{1}{2\pi }\int_{-\infty }^{+\infty }\left[D_{n}^{\left(1\right)}(k_{z})H_{n}^{\left(1\right)}(k_{r}r)e^{ik_{z}z}+D_{n}^{\left(2\right)}(k_{z})H_{n}^{\left(2\right)}(k_{r}r)e^{ik_{z}z}\right]dk_{z}$$
where $H_{n}^{\left(1\right)}$ and $H_{n}^{\left(2\right)}$ are two types of Hankel function, $D_{n}^{\left(1\right)}(k_{z})$ and $D_{n}^{\left(2\right)}(k_{z})$ are unknowns. In the case of out-of-column acoustic radiation, all sources of radiation are included in the cylinder r=a. There is no cylindrical wave converging inward, only the sound waves propagating outward are considered, so in Equation (10), $D_{n}^{\left(2\right)}(k_{z})=0$. Therefore, the solution to the problem of acoustic radiation outside the cylinder is:(11)$$p(r,\theta ,z)=\sum_{n=-\infty }^{+\infty }e^{in\theta }\frac{1}{2\pi }\int_{-\infty }^{+\infty }\left[D_{n}^{\left(1\right)}(k_{z})H_{n}^{\left(1\right)}(k_{r}r)e^{ik_{z}z}\right]dk_{z}$$

Elementary wave functions in the spatial frequency domain determined by wave-number vector $K_{m}=(n,k_{z})$ is:(12)$$\Phi_{Km}(r,\theta ,z)=e^{in\theta }e^{ik_{z}z}\frac{H_{n}^{\left(1\right)}(k_{r}r)}{H_{n}^{\left(1\right)}(k_{r}a)}$$

In the cylindrical SONAH, for Equation (12), the radius r of a certain cylindrical measurement surface is determined, and the radius r of the cylinder sound source is also determined. Thus, for an actual cylindrical sound source, its axial coordinates z and circumferential coordinates θ vary in the cylindrical coordinate system, so there are two variables in the expression of element cylindrical waves. However, for a conical source, the radius varies with the axial distance but the cone angle of measurement surface or sound source surface is fixed. For an actual conical source, given the size of the cone angle, a certain point on the cone can be only determined by the axial and circumferential coordinates. Although the radial size of the conical sound source is also variable, it can be expressed by the geometric relationship with the axial coordinates and cone angle. In view of the above analysis, considering the similarity between the conical surface and cylindrical surface, cylindrical SONAH is extended to the conical surface. To further illustrate its implementation process, the conical near-field acoustic holographic measurement scheme is shown in Figure 2, and the measurement surface is expanded as shown on the right. The outermost conical surface is the holographic measurement surface, and the length along the conic bus is *L*. The conical surface with the radius $r_{s}$ of the upper surface is the reconstruction surface, and the sound sources are all contained in the conical surface with the radius $r_{a}$ of the upper surface.

The front view of the measurement surface is shown in Figure 3. The conical Angle remains unchanged over all conical surfaces, and the source surface is located at the dotted line. Therefore, the cylindrical invariant in Equation (12) can be expressed as the cone angle of the conical surface, and Equation (12) can be changed into the following form:(13)$$\Phi_{k}^{\beta }((z\times \tan\frac{\beta }{2}+rh_{1}),\theta ,z)=e^{in\theta }e^{ik_{z}z}\frac{H_{n}^{\left(1\right)}(k_{r}z\times \tan\frac{\beta }{2}+rh_{1})}{H_{n}^{\left(1\right)}(k_{r}z\times \tan\frac{\beta }{2})}$$

If the integral operation is discretized, the sound pressure on the holographic and reconstructed conical surface can be expressed as:(14)$$\begin{array}{l} p((z\times \tan\frac{\beta }{2}+rh_{1}),\theta ,z)\approx \frac{1}{2\pi }\sum_{m=1}^{M}P_{n}(a,K_{m}^{\beta })\Phi_{Km}^{\beta }(r_{H},\theta ,z)\\ =\frac{1}{2\pi }\sum_{m=1}^{M}P_{n}(k_{r}z\times \tan\frac{\beta }{2},K_{m}^{\beta })e^{in\theta }e^{ik_{z}z}\frac{H_{n}^{\left(1\right)}(k_{r}z\times \tan\frac{\beta }{2}+rh_{1})}{H_{n}^{\left(1\right)}(k_{r}z\times \tan\frac{\beta }{2})} \end{array}$$

Similarly, according to the superposition principle of wave field, the characteristic surface wave of the same wavenumber vector has the superposition property. Then, for the reconstruction of the conical surface, the characteristic surface wave at any point $r_{S}=(r_{S},\theta ,z)$ with the wavenumber vector $K_{m}$ can be obtained by the superposition of the characteristic surface wave with the same wavenumber vector $K_{m}$ at all points $r_{Hn}=(r_{H},\theta_{n},z_{n})$ (here $r_{H}=z_{n}\times \tan\beta +rh_{1}$) on the holographic surface.
(15)$$\Phi_{Km}^{\beta }(r_{S})=\sum_{n=1}^{N}C_{n}^{\beta }(r_{S})\Phi_{Km}^{\beta }(r_{Hn}),m=1,2,\cdots ,M$$
where $r_{Hn}=(r_{H},\theta_{n},z_{n})(n=1,2,\cdots ,N)$ is a measuring point of sound pressure on the holographic cone, M is the number of characteristic surface waves contained in the complex pressure on the reconstructed cone and holographic cone, $C_{n}^{\beta }(r_{S})$ is the superposition coefficient.

Substitute Equation (15) into Equation (14) to obtain:(16)$$p(r_{S},\theta ,z)\approx \frac{1}{2\pi }\sum_{m=1}^{M}P_{n}(a,K_{m}^{\beta })\Phi_{Km}^{\beta }(r_{H},\theta ,z)=\sum_{n=1}^{N}C_{n}^{\beta }(r_{S})p(r_{Hn})$$

A system of linear equations can be determined by Equation (15):(17)$$\alpha_{\beta }(r_{S})=\left[\begin{array}{c} \Phi_{K1}^{\beta }(r_{S})\\ \Phi_{K1}^{\beta }(r_{S})\\ \vdots \\ \Phi_{K1}^{\beta }(r_{S}) \end{array}\right]$$
(18)$$C_{n}^{\beta }(r_{S})=\left[\begin{array}{c} C_{1}^{\beta }(r_{S})\\ \vdots \\ \vdots \\ C_{N}^{\beta }(r_{S}) \end{array}\right]$$
(19)$$A_{\beta }=\left[\begin{array}{cccc} \Phi_{K1}^{\beta }(r_{H1}) & \Phi_{K1}^{\beta }(r_{H2}) & \cdots & \Phi_{K1}^{\beta }(r_{HN})\\ \Phi_{K2}^{\beta }(r_{H1}) & \Phi_{K2}^{\beta }(r_{H2}) & \cdots & \Phi_{K2}^{\beta }(r_{{}_{HN}})\\ \vdots & \vdots & \ddots & \vdots \\ \Phi_{KM}^{\beta }(r_{H1}) & \Phi_{KM}^{\beta }(r_{H2}) & \cdots & \Phi_{KM}^{\beta }(r_{HN}) \end{array}\right]$$To make the solution unique, M≥N is required. Moreover, the Equation can be expressed as:(20)$$\alpha_{\beta }(r_{S})=A_{\beta }C_{n}^{\beta }(r_{S})$$

To obtain the coefficient matrix $C_{n}^{\beta }(r_{S})$, first, for the finite subset of elementary cylinder wave $\Phi_{Km}^{\beta }(r_{S})$, Equation (15) gives an optimal estimate. Similarly, for Equation (16), the optimally estimated sound pressure of reconstruction surface can also be given by appropriate weighted processing. Next, the regularization method is used to suppress the influence of the small-scale evanescent waves, and the regularization of the above Equation is obtained as:(21)$$C_{n}^{\beta }(r_{S})={\left(A_{\beta }^{H}A_{\beta }+\theta^{2}I\right)}^{-1}A_{\beta }^{H}\alpha_{\beta }(r_{S})$$
where $A_{\beta }^{H}$ is the conjugate transposed matrix of $A_{\beta }$, θ is the regularization parameter that acts as a filter, I is the unit diagonal array.

### 2.2. Theories of the Combined Regularization Method

Statistically optimal near-field acoustic holography is an acoustic inverse problem. The ill-posed problem is very sensitive to measurement errors, but the actual measurement data contains errors inevitably, so it cannot be solved directly by conventional methods. However, it does not mean that a meaningful solution cannot be obtained. A combined regularization method based on TSVD and Tikhonov is proposed to obtain a stable and meaningful solution to the inverse problem. Before introducing the combined regularization method, it is necessary to have a detailed overview of the role of TSVD and Tikhonov regularization in sound field reconstruction:

#### 2.2.1. *Truncated Singular Value Decomposition (TSVD)*

The main idea of TSVD is to find a good matrix $A_{k}$ to make it better approximate the matrix A under the 2-norm. To filter out the contribution in solution $\frac{u_{i}^{T}b}{\sigma_{i}}v_{i}$ of small $\sigma_{i}$, corresponding $f_{i}$ can be 0 and the others are 1. The solution of the Equation b=Ax can be written as:(22)$$x_{\mathrm{reg}}=\sum_{i=1}^{n}f_{i}\frac{u_{i}^{T}b}{\sigma_{i}}v_{i},\mathrm{if}L=I_{n}$$

In this method, all singular values smaller than the threshold value are discarded, which reduces the condition number of the acoustic transfer matrix and improves its ill-posed property. However, the truncated small singular values contain the high spatial frequency information of the sound field. Although it can suppress the amplification effect of noise and other errors hidden in the high-frequency information in the reconstruction process, the lost high-frequency detail information of the sound field reduces the accuracy of reconstruction. In addition, for Equation (22), the key lies in the selection of truncation points. The selection of a very small truncation point will result in losing a large amount of high-frequency components in the sound field, while a very large truncation point would not be able to suppress the influence of noise. The truncation point is determined by the contribution rate CR [34] of singular values in this paper:(23)$$CR=\left(\sigma_{j}/\sum_{j=1}^{n}\sigma_{j}\right)\times 100\%$$
where $\sigma_{j}$ is the singular value. In engineering practice, the contribution rate of singular values is usually 1~5%.

#### 2.2.2. *Tikhonov Regularization*

Another way of suppressing the effect of the right-end error is to add a constraint when solving the original discrete problem, which limits the “size” of the solution (measured by the appropriate norm) to make the solution smoother. Therefore, the problem can be described as the following optimization problem:(24)$$\underset{x}{\min}\left\{{\left\|Ax-b\right\|}_{2}^{2}+\lambda^{2}\Omega {\left(x\right)}^{2}\right\}$$
where λ is the regularization parameter, Ω(x) is a smooth norm. This regularization method is known as Tikhonov regularization. A balance between the residual norm and the “size” of the solution is controlled by λ. If λ, Equation (24) degenerates into a least square problem, the solution is not regularized. When solving Equation (24), by taking a discrete smooth norm $\Omega (x)={\left\|Lx\right\|}_{2}$, the Equation (24) becomes:(25)$$\underset{x}{\min}\left\{{\left\|Ax-b\right\|}_{2}^{2}+\lambda^{2}{\left\|Lx\right\|}_{2}^{2}\right\}$$
where L is a real regularization matrix of p×n dimensions, usually p≤n. The Equation (25) is called the general form of Tikhonov. When $L=I_{n}$, it is called the standard form of Tikhonov. Equation (24) can be expressed as the following Tikhonov problem:(26)$$(A^{H}A+\lambda^{2}L^{T}L)x_{\mathrm{reg}}=A^{H}b$$Its solution is:(27)$$x_{\mathrm{reg}}={\left(A^{H}A+\lambda^{2}L^{T}L\right)}^{-1}A^{H}b$$If $L=I_{n}$, the Tikhonov regularization can be written as follows through the TSVD:(28)$$x_{\mathrm{reg}}=\sum_{i=1}^{n}\frac{\sigma_{i}^{2}}{\sigma_{i}^{2}+\lambda^{2}}\frac{u_{i}^{T}b}{\sigma_{i}}v_{i},\mathrm{if}L=I_{n}$$

Therefore, the filtering factor of the Tikhonov regularization is $f_{i}=\sigma_{i}^{2}/\left(\sigma_{i}^{2}+\lambda^{2}\right)$.

As can be seen from Equation (28), Tikhonov regularization revised all singular values of the acoustic transfer matrix, not only the smaller singular values corresponding to the evanescent wave of high spatial frequency, but also the larger singular values corresponding to the propagation wave of low spatial frequency which does not contain the noise. In this way, low-frequency information components without high-frequency noise will be distorted, which will affect the accuracy of sound field reconstruction. Moreover, the improper selection of regularization parameters will cause over-filtering and under-regularization.

#### 2.2.3. *Combined Regularization*
*Method*

As can be seen from Equations (22) and (28), the difference between the two regularization methods is the number of corrections for singular values. Based on this, the combined regularization method is proposed, where the singular values of the acoustic transfer matrix are treated by different regularization methods. Figure 4 shows the schematic diagram of the combined regularization method. Firstly, the truncation point in TSVD is selected by the appropriate contribution rate to truncate the singular values of the matrix. Then, because the singular values before the truncation point correspond to the low-spatial frequency components of the sound field and which do not contain high-frequency noise, they are not processed. Finally, since the small singular values after the truncation point correspond to the high-spatial frequency components of the sound field containing high-frequency noise, measuring errors such as noise will be amplified with the reconstruction of high-frequency evanescent waves. Therefore, it is necessary to use the Tikhonov regularization to modify the small singular values.

Based on the above analysis, the solution of the combined optimized regular method can be obtained as:(29)$$x_{T\&\mathrm{Tik}}=\sum_{i=1}^{k}\frac{u_{i}^{T}b}{\sigma_{i}}v_{i}+\sum_{i=k+1}^{n}\frac{\sigma_{i}^{2}}{\sigma_{i}^{2}+\lambda^{2}}\frac{u_{i}^{T}b}{\sigma_{i}}v_{i}$$

From the above Equation, the combined regularization method also has the problem of selection of regularization parameters *k* in TSVD and λ in Tikhonov. The truncation point *k* is determined by the method of Equation (23). In the latter term of Equation (29), the regularization parameter λ in Tikhonov is the singular value corresponding to the truncation point, which ensures that there is only one undetermined parameter in the combined regularization and that all singular values contribute to the solution. It not only avoids the loss of high-frequency components of the sound field caused by the TSVD method, but also restrains the influence of measuring errors such as noise in the reconstruction accuracy. Based on the above analysis, the final solution obtained by combined regularization method is:(30)$$x_{T\&\mathrm{Tik}}=\sum_{i=1}^{k}\frac{u_{i}^{T}b}{\sigma_{i}}v_{i}+\sum_{i=k+1}^{n}\frac{\sigma_{i}^{2}}{\sigma_{i}^{2}+\sigma_{k}^{2}}\frac{u_{i}^{T}b}{\sigma_{i}}v_{i}$$
where $x_{T\&\mathrm{Tik}}$ is the solution of combined regularization, *k* is the truncation point.

## 3. Numerical Case Studies

### 3.1. Quantitative Study on the Influence of Cone Angle

In cylindrical SONAH, when the axial size of a cylindrical sound source is fixed, the radiated sound field is affected by the radius of the radiation source. However, in a conical source, when the axial size is constant, the radiated sound field of the source is affected by both cone angle and radius. Therefore, it is necessary to study the influence of cone angle on the accuracy of acoustic field reconstruction. Conical SONAH is used to explore the reconstruction effect at different cone angles.

Simulation Settings are as follows: The line array of pulsating balls with gradually increasing radius placed along (x,y)=(0,0), spanning from z=0 to 1.0 m are adopted to simulate the conical sound source. The distance among them is kept much lesser than half of the sound wavelength and the frequency is 300 Hz. The radius of the small bore of the measurement surface is 0.1 m, the axial dimension is 1.0 m, the circumference interval is 18°, the axial interval to be 0.05 m, and the reconstruction distance (the distance from the measurement surface to the reconstruction surface) is 0.05 m. During the simulation process, random noise with a signal-to-noise ratio of 20 dB is applied to the measurement surface.

The amplitude error of a certain reconstructed point is defined as [34]:(31)$$L_{e}=\sqrt{\frac{\sum_{i}{\left(\left|P_{si}\right|-\left|P_{ri}\right|\right)}^{2}}{\sum_{i}{\left|P_{ri}\right|}^{2}}}\times 100\%$$where *L*_*e*_ is the total relative error, $P_{si}$ is the sound pressure at a point of reconstruction surface, $P_{ri}$ is the theoretical sound pressure at a point of the reconstruction surface.

Based on the sound pressure on the measurement/holographic surface, the proposed conical SONAH is used to reconstruct the sound field on the reconstruction surface. To observe the degree of agreement between the reconstructed values and theoretical values on the reconstruction plane more clearly, Figure 5 depicts the distribution of the reconstructed and theoretical sound pressure amplitude along θ=0 on the reconstruction surface under the condition of different cone angles. It can be seen that when the cone angles are 15° and 30°, the reconstructed sound pressure amplitude is in good agreement with the theoretical amplitude except for the edge points of the conical surface. When the cone angle increases to 45° and 60°, the reconstructed values of the edge points differ greatly from the theoretical values, and the reconstructed values of other points are also not in good agreement with the theoretical values.

Analyzing the reason for this phenomenon, for the conical shell structure sound source, the holographic measurement of sound pressure is limited in the axial direction in the actual application process, so there is a discontinuity at the edge of the measurement aperture, causing the leakage of spatial frequency. Hence, the reconstructed value at the edge of the reconstruction surface is diverged and becomes meaningless. Furthermore, Figure 6 shows the amplitude variation rule of characteristic waves with radius in the conical spatial domain. As it can be seen, when the radius of the measurement surface decreases, the amplitude of the characteristic surface wave also decreases, and the number of smaller singular values of the matrix increases. At the same time, the value of the superposition coefficient vector corresponding to the solved sound pressure of small bore is unstable, which leads to a larger reconstruction error in the small section of the cone than in the large bore.

To further illustrate the above conclusions, Figure 7 shows the total relative errors of reconstruction at different cone angles and at different frequencies. When the cone angle increases from 15° to 75°, the total relative error of reconstruction also increases. The reason for this is that the characteristic wave applicable to the conical sound source is obtained by the element cylindrical wave through the orthogonal coordinate transformation, and in the cylindrical wave function of Equation (12), its radius is constant and when θ is determined, the element cylindrical wave is only related to the axial coordinate. In Equation (13), the radius of a fixed cone varies. When θ is fixed, the characteristic wave of the conical SONAH is not only related to the axial coordinate but also affected by the radius. Moreover, with the increase in the cone angle, the radius variation range of the cone also increases, leading to a greater difference between the maximum and minimum singular values in the acoustic field transfer matrix, aggravating the ill-health of the transfer matrix, and enhancing the amplification effect of measurement errors such as noise in the reconstruction process, so that the reconstruction error increases.

Through the above investigations, the influence of cone angle on the accuracy of acoustic field reconstruction is obtained, which indicates that the proposed method is practical in engineering.

### 3.2. Comparative Study of Conical and Cylindrical SONAH

Conical SONAH is proposed in Section 2.1, then considering that cylindrical SONAH can also be used to directly reconstruct the sound field of the conical sound source, and Figure 8 shows a schematic diagram of the two methods. The direct reconstruction of cylindrical SONAH is based on cylindrical measurement, and the characteristic surface wave matrix of the cylindrical spatial domain is used to reconstruct the radiated sound field of the conical sound source. In comparison, the conical spatial domain characteristic surface wave matrix is used in conical SONAH, which is proposed in this paper to reconstruct the sound field by conical measurement.

Figure 9 and Figure 10 show the comparison of reconstruction effects between the two methods when the cone angle is 20° and 40° respectively. As it can be seen from Figure 9, when the cone angle is 20°, the proposed conical SONAH method can effectively reconstruct the sound field radiated by the cone sound source. The theoretical and reconstructed sound pressure values are in good agreement except for the edge points of the reconstruction surface. However, it is impossible to reconstruct the radiated sound field of the conical sound source by using cylindrical SONAH. Even in the central area of the reconstruction surface, the reconstructed value of each point is quite different from the theoretical value.

As it can be seen from Figure 10, in comparison to the cylindrical SONAH, when the cone angle increases up to 40°, the proposed conical SONAH method still has better reconstruction performance. Furthermore, Figure 11 shows the total relative errors of two methods in the whole reconstruction surface when the cone angle varies from 15° to 75°. For the conical sound sources with different cone angles, the overall relative errors of the proposed conical SONAH are smaller than that of the cylindrical SONAH, which can be reduced by about 15%. When the cone angle increases, the advantages of the proposed method can be highlighted, and the maximum reconstruction error can be reduced by about 30%.

Therefore, it can be seen from the above analysis that the proposed conical SONAH method can achieve a more effective reconstruction of the sound field radiated by the conical sound source than that of the cylindrical SONAH, indicating the effectiveness of the proposed method. When the cone Angle is 40°, Figure 12 shows that the overall relative error of the proposed method is about 25% higher than that of direct reconstruction at different SNRs and frequencies. Therefore, it is verified that the proposed method has higher accuracy and robustness under different reconstruction parameters.

### 3.3. Quantitative Study of Combined Regularization Method

Just like cylindrical SONAH, the conical SONAH method is an acoustic inverse problem, and its unfitness makes it difficult to obtain the real solution of the problem by standard numerical method. Therefore, it is necessary to use the corresponding regularization method to obtain the stable and meaningful solution of the inverse problem.

#### 3.3.1. *Ill-Condition Analysis of Acoustic Transfer Matrix*

As described in the previous section, the direct inverse method cannot be used to solve the ill-posed problem of large-number conditions of the acoustic field transfer matrix. Before introducing the regularization method, it is necessary to study the several important factors affecting the conditional number of transfer matrix in conical SONAH. It is required then to keep other simulation parameters the same as the ones presented in Section 3.2 and change the cone angle, measuring distance, and microphone spacing. Finally, the corresponding conditional number of the acoustic field transfer matrix of transfer matrix in conical SONAH can be calculated. The obtained results are shown in Figure 13, Figure 14 and Figure 15.

From the above figures, we can draw the following conclusions: when the size of cone angle and the microphone spacing changes, the condition number of the matrix does not change much and when the measuring distance increases, the condition number tends to increase. However, it is worth noting that no matter how the conditional number of the matrix changes, it is always above a large value. Therefore, the acoustic transfer matrix is seriously ill-conditioned and needs to be dealt with by the regularization method.

#### 3.3.2. *Comparative Study**of Different Regularization Methods*

Simulation settings are shown in Section 3.1, and the case of a cone angle of 30° is taken as an example to compare different regularization methods of the conical SONAH. Figure 16 shows the reconstruction effect of the Tikhonov regularization parameter selection method combined with the Hald criterion. When the SNR is 30 dB, the reconstructed value is in good agreement with the theory; when the SNR increases, the reconstruction effect becomes worse. To analyze the reasons, the Hald criterion selects regularization parameters according to the following equation:(32)$$\theta^{2}=(1+\frac{1}{{\left(2kd\right)}^{2}})\times 10^{\frac{-\mathrm{SNR}}{10}}$$
where SNR is the signal to noise ratio, d is the distance between measuring and reconstruction surface. As can be seen from Equation (32), when the signal-to-noise ratio increases, the regularization parameter decreases in order of magnitude of 10, which leads to the lack of regularization of the problem. Therefore, it is impossible to effectively deal with small singular values, and the measurement errors such as noise are amplified.

Further, Figure 17 shows the reconstruction effect of Tikhonov combined with the GCV regularization parameter selection method. It is also difficult to achieve effective sound field reconstruction, and the reconstructed error obtained from Equation (31) is 97.79%. Simulation experiments were carried out on this method many times, and the results showed that this method had the problem of too small regularization parameter selection and could not realize reconstruction under different SNR. When the reconstruction (SNR = 30 dB) can be achieved by the Hald criterion, the comparison with the regularization parameters shows that the regularization parameter obtained by GCV are about 10 times smaller. Therefore, Tikhonov combined with the GCV method has the problem of under-regularization and cannot effectively realize sound field reconstruction.

Based on the above analysis, the single regularization method cannot realize the effective reconstruction of the acoustic field. Therefore, a combined regularization method is proposed in Section 2.2.3. In this method, the key is a way of determining the truncation point *k*. Table 1 shows the influence of singular value contribution rate on reconstruction error. When the contribution rate of the selected singular values increases, the overall relative error of sound field reconstruction firstly decreases and then increases. The reason for this is that when the contribution rate is less than 4%, the number of truncated smaller singular values increases with the increase in contribution rate, which can more effectively suppress the amplification of reconstruction errors caused by noise. At the same time, the retained large singular values can contain most of the energy of the sound field, so the reconstruction errors tend to decline. When the contribution rate is more than 4%, the number of truncated small singular values increases with the increase in contribution rate. In this way, the influence of noise on the reconstruction can be suppressed, however, high-frequency evanescent wave energy used for reconstruction is lost, so the reconstructed error increases. Therefore, the singular value contribution rate is 4% in this paper.

On this basis, to illustrate the advantages of the combined optimization method, Figure 18 shows the reconstructed sound pressure amplitude of TSVD and the combined regularization method when the cone angle is 30°, in which the Hald criterion is used to select the regularization parameter. Compared with the TSVD method, the reconstruction values of the combined optimization method are in better agreement with theoretical values at all points on the reconstruction surface.

The reconstruction performance of the two methods is further studied under different cone angles, and the results are shown in Figure 19. At each cone angle, the overall relative error of the reconstruction by the combined regularization method is smaller than that by the TSVD method, and the maximum reduction is about 5%. The reason is analyzed as follows: in the TSVD method, some small singular values are discarded directly, which can suppress the amplification effect of measurement errors such as noise to some extent. However, detailed information of high-frequency evanescent waves in the sound field is also discarded, so that the energy of the sound field used for reconstruction is lost. In the combined regularization method, instead of directly discarding the small singular values after truncation, it is further filtered. In the reconstruction process, the high-frequency details in the sound field are not lost, and the influence of measurement errors such as noise amplified with evanescent waves in the reconstruction process on the reconstruction results is suppressed, so the reconstruction error is smaller.

All the above studies are conducted under the condition of a certain signal-to-noise ratio. Furthermore, Figure 20 shows the overall relative errors of the combined regularization method in the reconstruction of different signal-to-noise ratios, so as to study the stability of the combined optimization method in the reconstruction process. It can be seen that with the increase in SNR, the overall relative error of sound field reconstruction tends to decrease, but changes little. For different cone angles, when the signal-to-noise ratio decreases from 50 dB to 5 dB, the overall relative error of reconstruction only increases by about 2%, which fully demonstrates that the proposed combined optimization method can effectively suppress the influence of noise and other measuring errors on reconstruction results, reflecting the stability of its solution. 

## 4. Experimental Case Studies

### 4.1. Introduction of Experimental System

To evaluate the actual performance of the proposed combined method for large conical surfaces, a test bed is constructed as shown in Figure 21. The test bed is a combination structure of hemisphere, cylinder, and cones. The hemisphere and cones are connected with cylinders through interference fitting, which can be disassembled and assembled according to research requirements, as shown in Figure 22.

The test bench is mainly composed of a combination structure of a plate and shell, an eccentric block vibration motor, and a bracket. The submarine hull is simulated by the combined shell and shell structure, and the submarine power device, the vibration source, is simulated by two eccentric block vibration motors. The plate and shell combination structure mainly comprises a cylindrical shell, a reinforcing rib, and a horizontal plate, and the horizontal plate is used for installing the eccentric block vibration motor. A motor-supported vibration-damping structure is added to the horizontal plate to simulate the power equipment base of the submarine. The motor supporting vibration damping structure is composed of a supporting platform and a damping rubber, and the four supporting legs are elastically connected with the shell transverse plate through the damping rubber and the bolt. The four-shell rubber spring and two brackets are used to realize the elastic isolation of the shell-and-shell structure from the ground to simulate the suspension state of the submarine in the water. When the motor is turned on, the vibration generated by the vibration source is transmitted to the cylindrical shell through the supporting structure of the motor, and then to the cone and hemispherical body through the contact coupling mode of interference coordination among the three, to stimulate the vibration on the whole composite structure surface and thus radiate the sound field to the three-dimensional space.

### 4.2. Test Implementation

#### 4.2.1. *Sensor Arrangement and Signal Acquisition*

The microphone array consists of 16 sensors, whose parameters are shown in Table 2. The measurement method adopts a single reference source transfer function method. To reduce the reflection of the surrounding walls and simulate the free sound field as much as possible, this test was carried out in a semi-anechoic chamber environment. The measurement scheme is shown in Figure 23. The data acquisition system uses 64-channel HBM, which can collect analog signals output by various types of sensors and convert them into digital signals and input them into computer processing to obtain specific data results. At the same time, the calculated waveforms and values are displayed in real time to monitor physical quantity status.

#### 4.2.2. *Test Implementation Steps*

In the conical SONAH test, only the large motor shown in Figure 23 is turned on. The vibration generated by the motor radiates the sound field to the space through the support and the surface of the conical shell.

The microphones are fixed on the microphone bracket. The number of microphones and the distance between microphones can be adjusted. In this experiment, 16 microphones were used, with a distance of 10 cm. The bracket is kept parallel to the axis of the shell, 15 cm away from the surface of the shell, and rotates around the axis of the shell to form a conformal conical test surface with the surface of the shell, to measure the sound pressure on the cone at a certain distance from the surface of the shell. Then, the microphone is adjusted from the surface of the shell to measure the sound pressure on the surface of the conical shell. Phase information of the sound pressure on all measurement points is calculated from the cross-spectrum of the sound pressure signals on a fixed reference point #1 as shown in Figure 23. The specific steps of the experiment are as follows:(1)As shown in Figure 23, connect the experimental equipment, set relevant parameters, pre-sample and roughly analyze the measurement data, and ensure that all equipment and channel signals are good.(2)Start the large motor alone and slowly adjust the motor speed to 1800 r/min. At this time, the vibration generated by the motor is transmitted to the cylindrical shell through the motor support, and then to the cone through the connection between the cone and the cylindrical shell, causing the vibration of the conical shell and thus radiating sound field into space.(3)The microphone bracket is rotated from one side of the shell to the other side at a circumferential interval of 22.5° to collect data of each circumferential angle position. Since the experimental conditions are limited, the radiative sound pressure of the upper part of the conical shell is measured.(4)The measurement distance is adjusted to the position 0.01 m away from the shell surface, and the above steps are repeated to obtain the radiation sound pressure data of the conical shell surface under the same condition.

### 4.3. Analysis of Test Data

The experimental data is processed by conical SONAH, and the amplitude error of reconstruction is defined in Section 3.1. Figure 24 shows the sound pressure amplitude distribution obtained by actual measurement of the measurement surface and reconstruction surface. The sound pressure amplitude decreases with the decrease in the cone section. The reason is that the vibration is transmitted from the motor vibration inside the cylindrical shell to the conical shell through the connection between the cylinder and the large section of the cone, and the vibration energy gradually attenuates in the transfer process. Moreover, the sound pressure amplitude on both sides of the conical plane is larger than the middle position. This is because the two surfaces are connected by interference fit. Due to the slight deformation of the conical shell and its large stiffness, the contact surface at the connection is not uniform, and the contact area on both sides is large and the coupling is firm, so larger vibration can be transmitted.

Figure 25 shows the reconstructed sound pressure amplitude obtained by the cylindrical SONAH direct reconstruction method (Traditional method) and the conical SONAH with different regularization methods. As compared to Figure 24b, the sound pressure amplitude directly reconstructed by cylindrical SONAH is quite different from that obtained by actual measurement. In the conical SONAH method, among three regularization methods, the sound pressure distribution reconstructed by the combined optimization method is consistent with the actual measured values, which preliminarily indicates that the proposed method can achieve more efficient reconstruction.

Table 3 shows the relative errors of reconstructed maximum with different methods, and the calculation methods are as follows:(33)$$e_{\max}=\left|(p_{\mathrm{true}}-p_{\mathrm{rec}})/p_{\mathrm{true}}\right|$$
where $e_{\max}$ is the relative error of reconstructed maximum, $p_{\mathrm{true}}$ is the measuring sound pressure, $p_{\mathrm{rec}}$ is the reconstructed sound pressure.

The reconstruction errors of the conical SONAH method are lesser than that of cylindrical SONAH. Moreover, it can be found that in the conical SONAH, the relative errors of the reconstruction maximum of existing regularization methods are greater than that of the combined regularization method proposed in this paper. As compared to cylindrical SONAH, the reconstruction error of conical SONAH with the combined regularization method proposed in this paper can be reduced by 23.56%.

In the above discussion, the reconstruction effect of the two methods is only roughly observed as a whole, and the relative error is only compared from the maximum value of reconstruction. To further explore the reconstruction effect on the entire reconstruction surface, the reconstruction surface is divided into different areas as illustrated in Figure 26, which expands from the center to the edge, so as to see the overall relative errors of the reconstruction of different areas by the two methods. Based on the above discussion, only conical SONAH with the combined regularization method is compared with the traditional cylindrical SONAH.

Figure 27 shows the reconstruction errors of the two methods in different regions at different frequencies. Firstly, for the two methods, the overall relative error of reconstruction also increases when the reconstruction region expands from the central region to the edge at different frequencies, which is consistent with the results in Section 3.3. On one hand, the actual measurement can only be carried out on the finite surface in the axial direction of the conical shell structure, so that the measurement signal is discontinuous and there is spatial frequency leakage, which leads to the increase in reconstruction error on the edge nodes of the reconstruction surface while on the other hand, it can be seen from the change rule of characteristic surface waves with the actual cone sound source radius in Figure 28, that smaller the measuring radius is, smaller the amplitude of characteristic surface waves will be, and the number of smaller singular values corresponding to the transfer matrix will increase. Therefore, when the conical surface with a small section is reconstructed, the small singular values will magnify the measurement errors such as noise, resulting in a larger reconstruction error at the small section of the conical surface.

Furthermore, the proposed methods have smaller reconstruction errors at different frequencies in different regions of the reconstruction surface. In each region, the error reduction is up to about 15%. The reason is as compared to the direct reconstruction method of cylindrical SONAH, the proposed conical SONAH constructs characteristic surface waves in the spatial domain of the conical surface through cylindrical and conical coordinate transformation, and then express the spatial sound pressure as the superposition of these characteristic surface waves to construct the acoustic transfer matrix, and finally realize the reconstruction of the sound field radiated by the cone sound source. While in the cylindrical SONAH, the construction of the acoustic transfer matrix is realized through element cylindrical waves. However, for cone sound sources, its radius varies with the axial position. Moreover, it is obvious from Figure 28 that cylindrical waves with a fixed radius cannot correctly represent this change rule, causing the greater reconstruction error. Therefore, the proposed conical SONAH has more obvious advantages than the direct reconstruction method of cylindrical SONAH.

In the proposed conical SONAH technology, a combined regularization method is proposed according to the characteristics of conical sound sources. In this method, the large singular values corresponding to the propagation waves without measurement noise are not processed to ensure the main energy acquisition of the sound field and avoid the distortion caused by regularization processing. The smaller singular values corresponding to the evanescent wave components with high spatial frequency are dealt with by Tikhonov regularization, so as to suppress the phenomenon that measurement errors such as noise are amplified with evanescent wave reconstruction, which further improves the reconstruction accuracy of the proposed method.

Based on the above experimental verification and result analysis, the conical SONAH proposed in this paper can effectively realize the reconstruction of the radiated sound field of the conical sound source at different frequencies and has higher reconstruction accuracy than the direct reconstruction method of cylindrical SONAH. It is worth noting to see that no parametric optimization of the hologram array or selection of characteristic wave function bases is performed in these simulations and experiments. It is possible to implement the conical SONAH method more successfully by altering the numerical measurement parameters, such as the distance between the hologram and sources, sensor density in the hologram, or by the inclusion of more wave functions.

## 5. Conclusions

In this paper, based on cylindrical SONAH, a conical SONAH with combined regularization method is proposed for the acoustic field reconstruction of conical sound sources through cylinder-cone coordinate transformation method to stabilize the solution of the acoustic inverse problem. The main conclusions of this study can be summarized as:

(1)In the theoretical framework, based on the cylindrical SONAH, the elementary cylindrical wave is transformed into the characteristic elementary wave suitable for the radiated sound field of the conical surface through the cylinder-cone orthogonal coordinate transformation method, and the matrix of characteristic elementary wave in the conical spatial domain is formed. Then, the sound pressure is expressed as the superposition of this characteristic elementary wave. Next, by incorporating the advantages of TSVD and Tikhonov, a combined regularization method is proposed to obtain a stable and meaningful solution to the inverse problem so that the superposition coefficient is solved according to the principle of superposition of wave field. Lastly, the reconstructed conical pressure is expressed as a linear superposition of the holographic conical pressure.(2)Simulation investigations demonstrated that the proposed conical SONAH can reconstruct the sound field of conical sound source efficiently. When the cone angle increases gradually, the reconstruction error of the sound field will increase accordingly at different frequencies. As compared to cylindrical SONAH, the proposed conical SONAH has better reconstruction accuracy under different cone angles, and the relative reconstruction error can be reduced by about 15%. As compared to the standalone Tikhonov regularization combined with GCV and TSVD method, the reconstruction error of the proposed combined method is reduced by 90% and about 5%. Therefore, the proposed combined regularization method is more accurate and stable.(3)The effectiveness of the modified method was validated through a set of experiments, and the results verified the better reconstruction performance of the modified method in comparison to the direct reconstruction method of cylindrical SONAH. It is difficult to achieve the effective reconstruction of the sound field with a single regularization method, while the proposed combination regularization method based on TSVD and Tikhonov can effectively solve the acoustic inverse problem to obtain high reconstruction accuracy. The relative error of the largest value is 11.14%, while that of the traditional method is 34.70%. Furthermore, the overall relative errors of different regions of the reconstruction surface are reduced by 10–15% with different frequencies. Therefore, the proposed method can effectively reconstruct the sound field radiated by the conical source. This study can provide a reference for the engineering application of conical SONAH and can benefit vibration and noise monitoring, reduction, and control for mechanical systems.

## Figures and Tables

**Figure 1 sensors-21-07150-f001:**
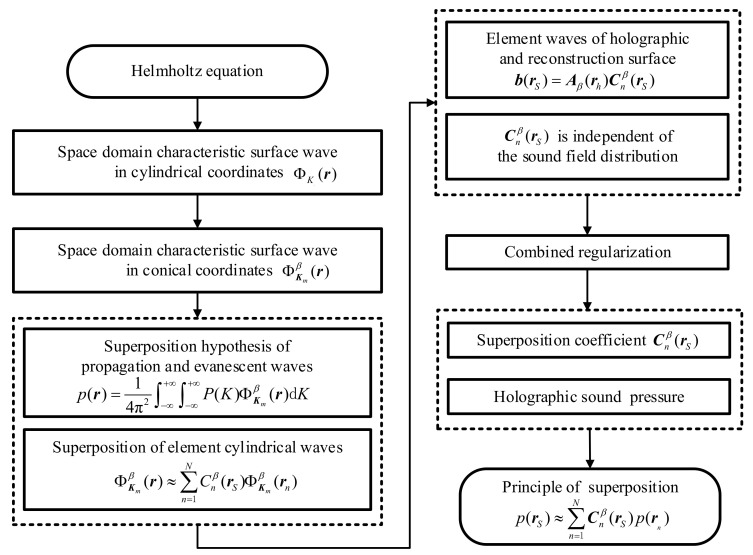
Implementation process of conical SONAH.

**Figure 2 sensors-21-07150-f002:**
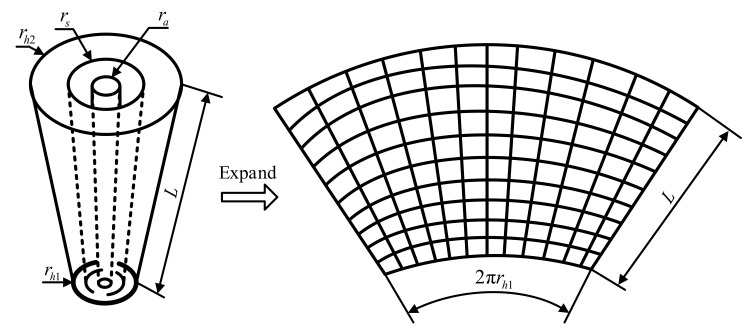
Diagram of conical surface measurement.

**Figure 3 sensors-21-07150-f003:**
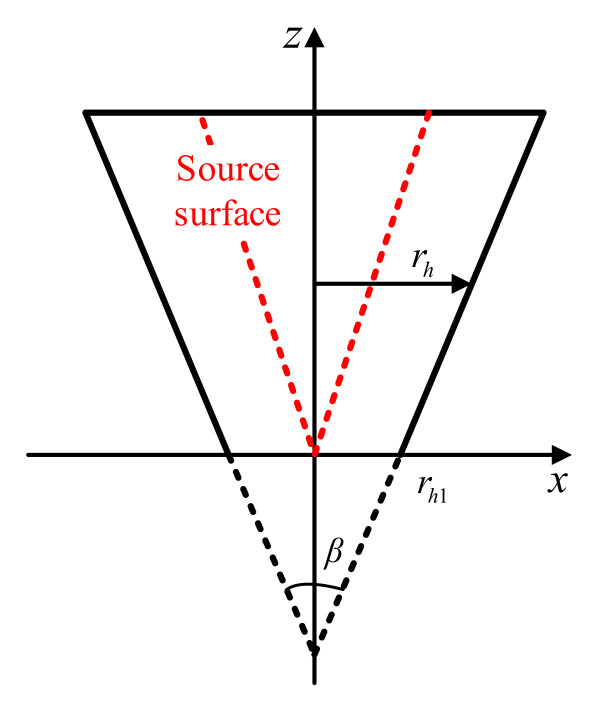
Front view of the measurement surface.

**Figure 4 sensors-21-07150-f004:**
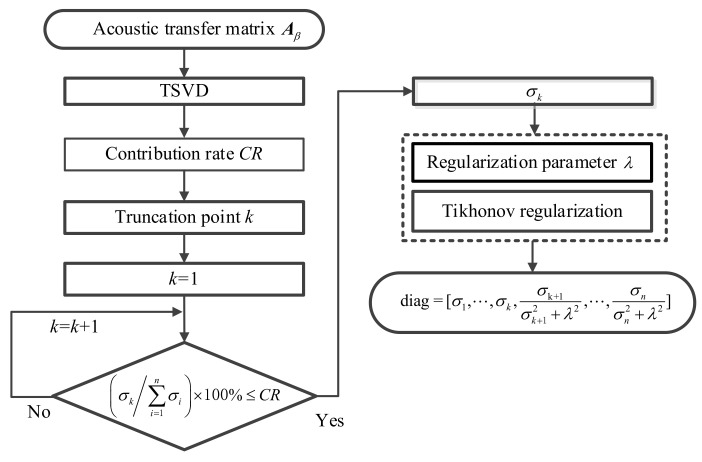
Schematic diagram of combined regularization method.

**Figure 5 sensors-21-07150-f005:**
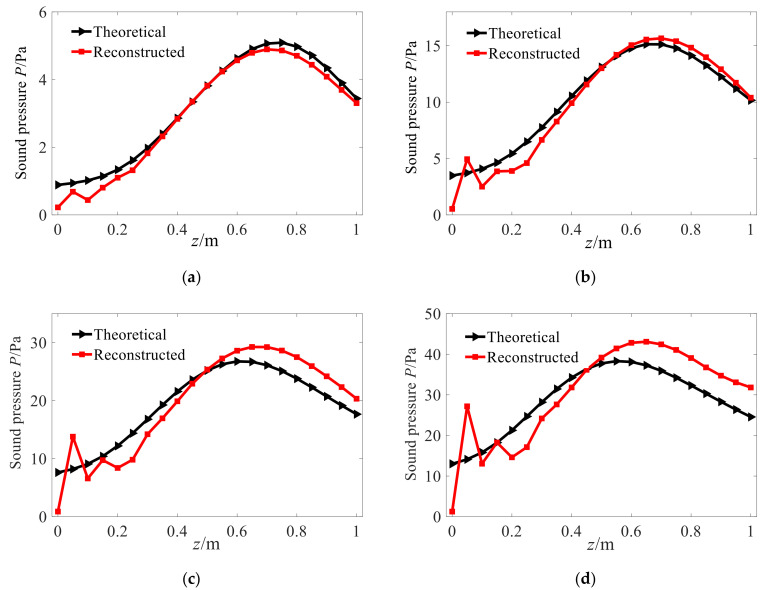
Pressure amplitude along θ=0. (**a**) Conical angle with 15°. (**b**) Conical angle with 30°. (**c**) Conical angle with 45°. (**d**) Conical angle with 60°.

**Figure 6 sensors-21-07150-f006:**
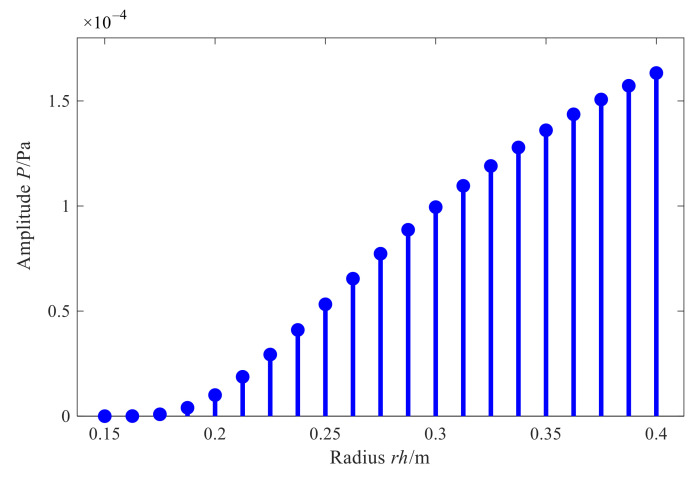
Amplitude variation of $\Phi_{Km}^{\beta }(r)$.

**Figure 7 sensors-21-07150-f007:**
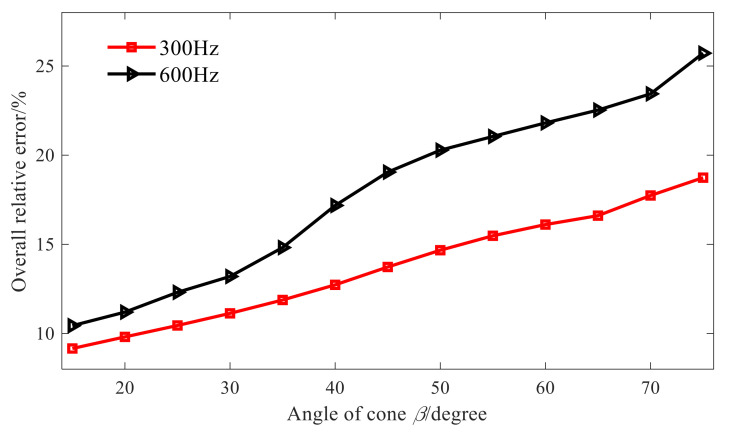
Overall errors of different angles.

**Figure 8 sensors-21-07150-f008:**
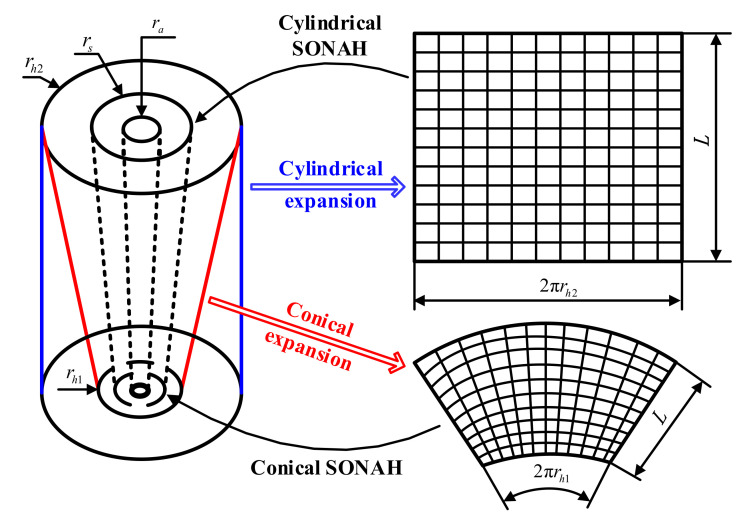
Schematic diagram of the two methods.

**Figure 9 sensors-21-07150-f009:**
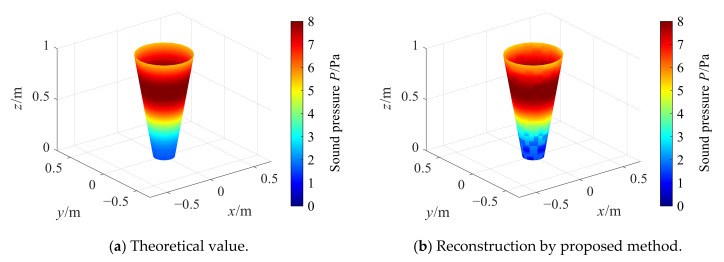
Reconstruction with cone angle 20°.

**Figure 10 sensors-21-07150-f010:**
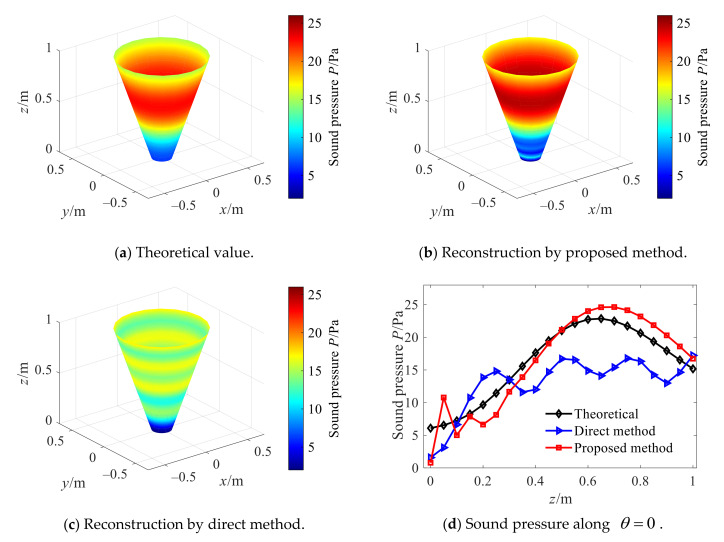
Reconstruction with cone angle 40°.

**Figure 11 sensors-21-07150-f011:**
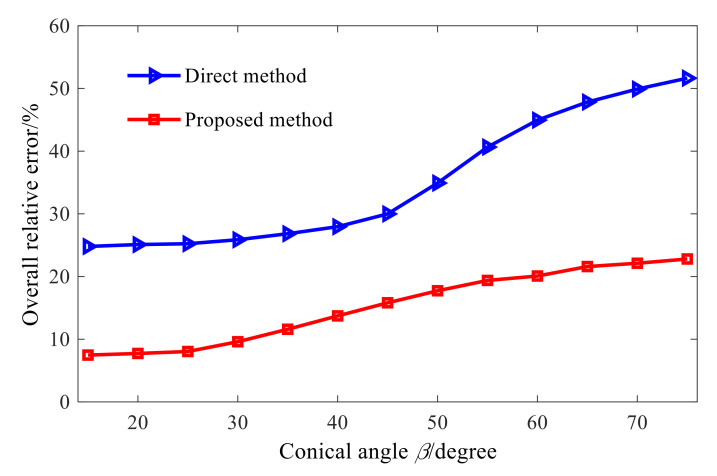
Overall relative errors of the two methods.

**Figure 12 sensors-21-07150-f012:**
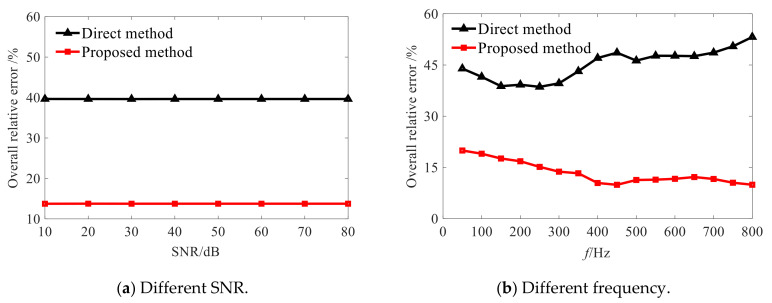
Overall relative errors of the two methods with 40° cone angle.

**Figure 13 sensors-21-07150-f013:**
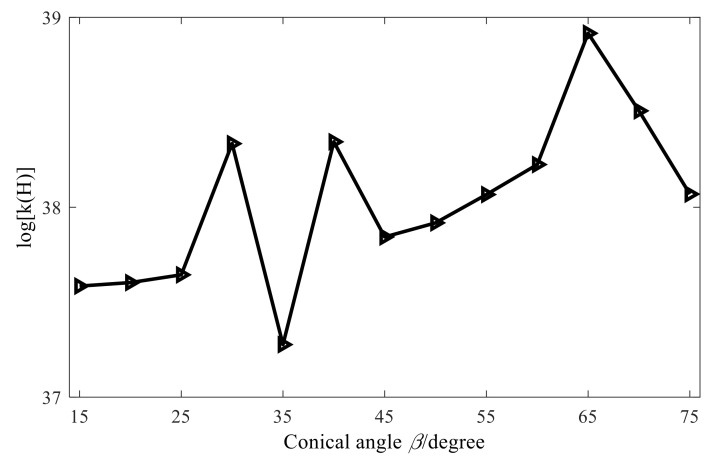
Influence of cone angle on conditional number.

**Figure 14 sensors-21-07150-f014:**
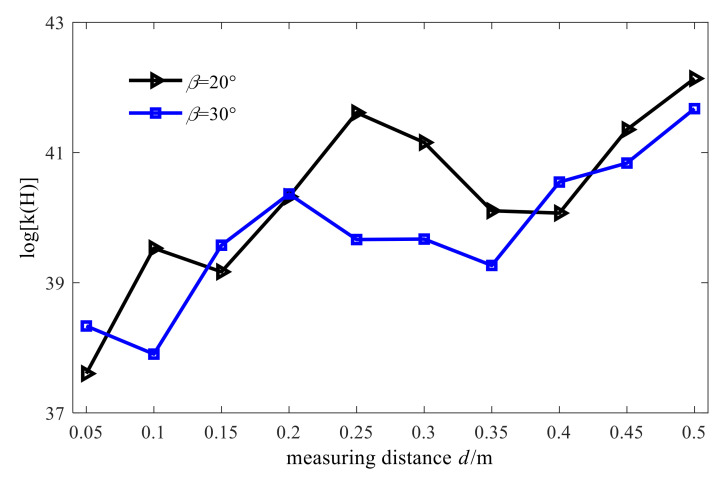
Influence of measuring distance on conditional number.

**Figure 15 sensors-21-07150-f015:**
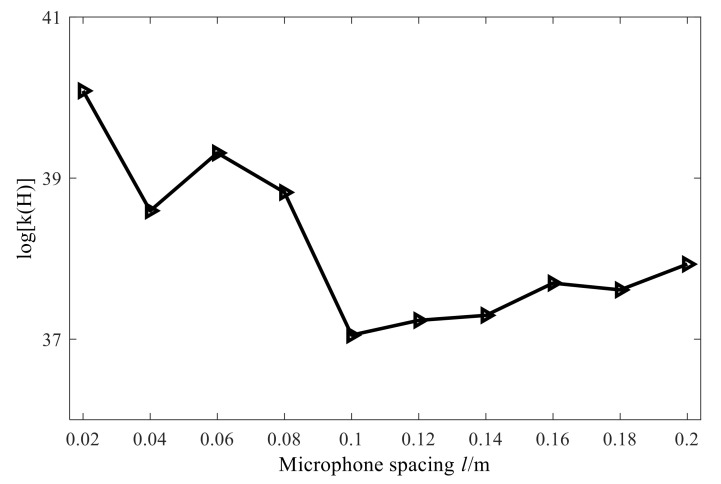
Influence of microphone spacing on conditional number.

**Figure 16 sensors-21-07150-f016:**
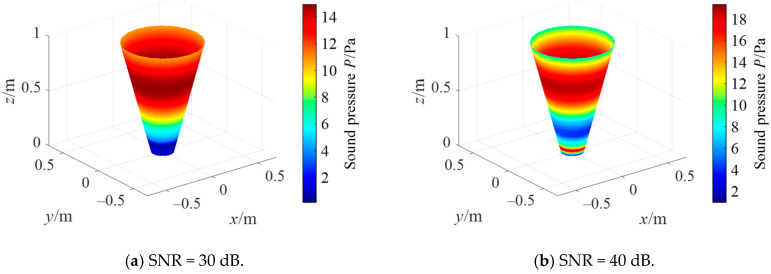
Reconstruction by Tikhonov with Hald criterion.

**Figure 17 sensors-21-07150-f017:**
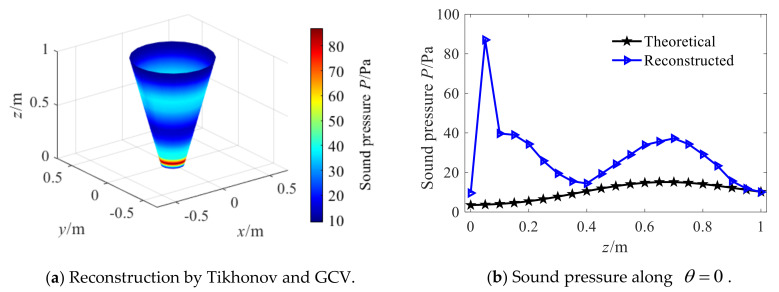
Reconstruction by Tikhonov and GCV.

**Figure 18 sensors-21-07150-f018:**
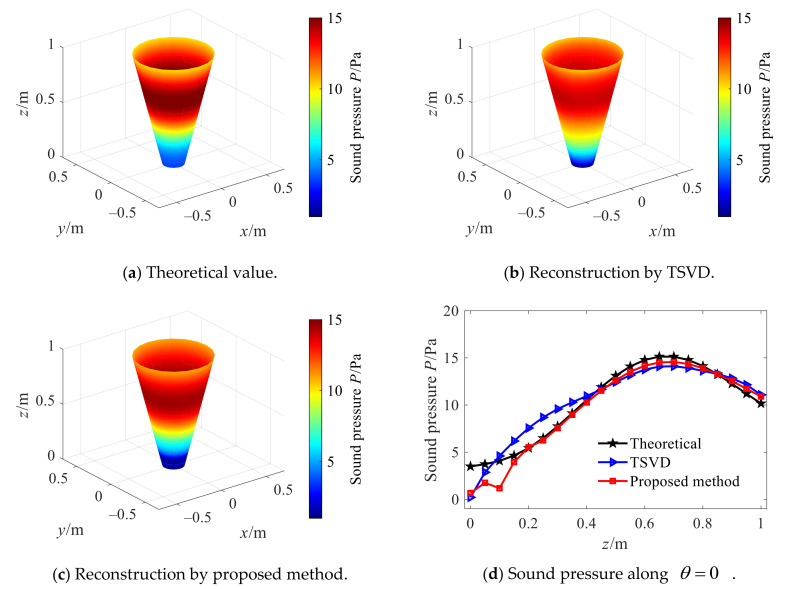
Reconstruction by TSVD and proposed method.

**Figure 19 sensors-21-07150-f019:**
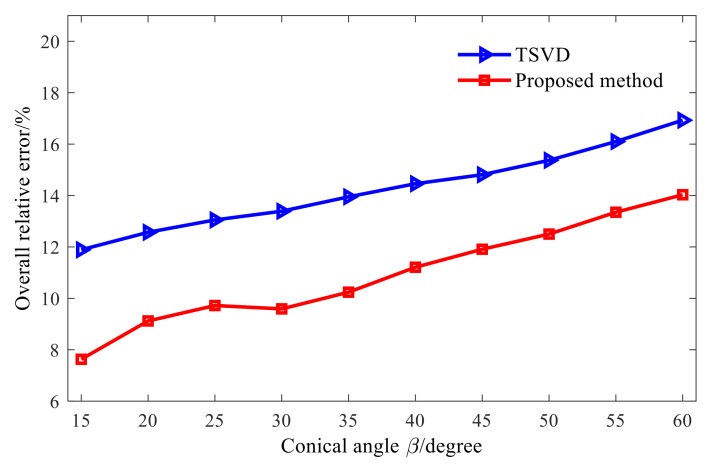
Reconstruction by the two methods under different cone angles.

**Figure 20 sensors-21-07150-f020:**
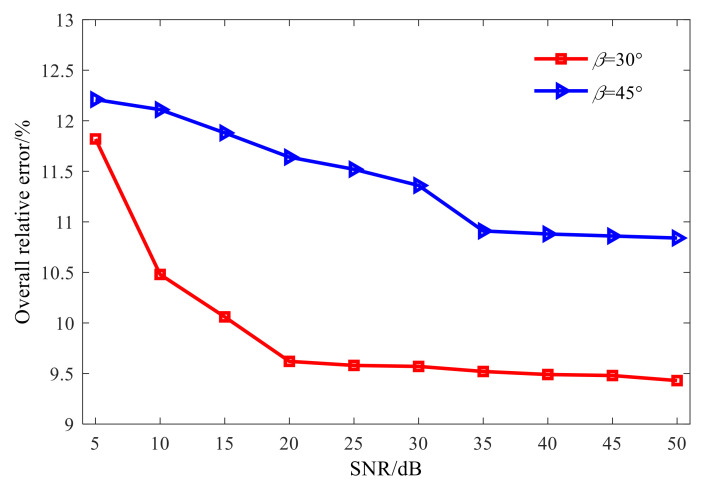
Overall relative errors of the proposed method.

**Figure 21 sensors-21-07150-f021:**
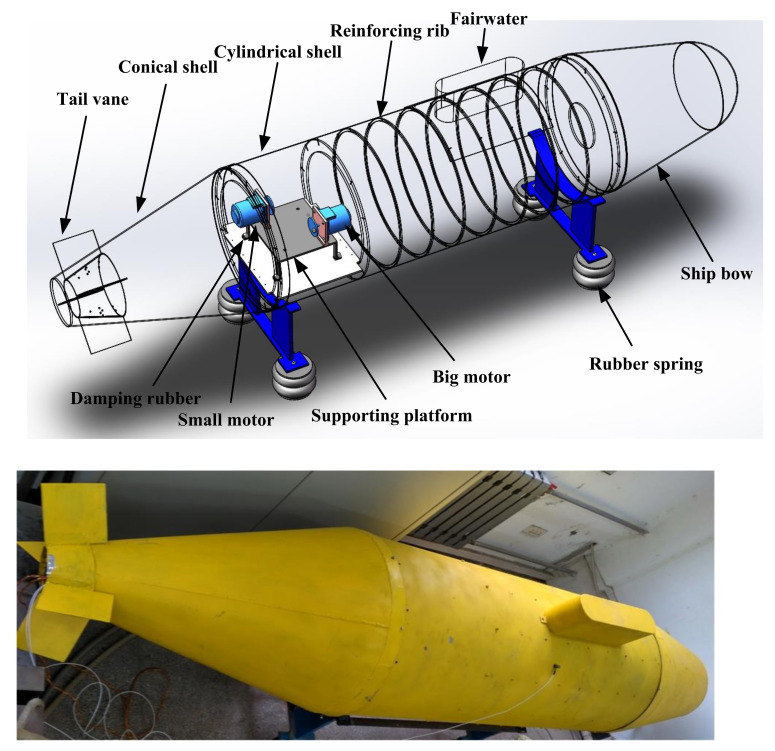
Composite structure shell test bed.

**Figure 22 sensors-21-07150-f022:**
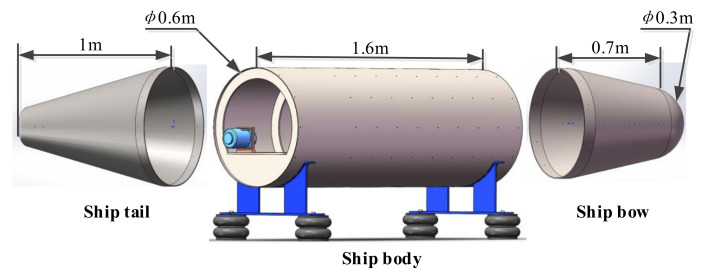
Structure of each part of the test bed.

**Figure 23 sensors-21-07150-f023:**
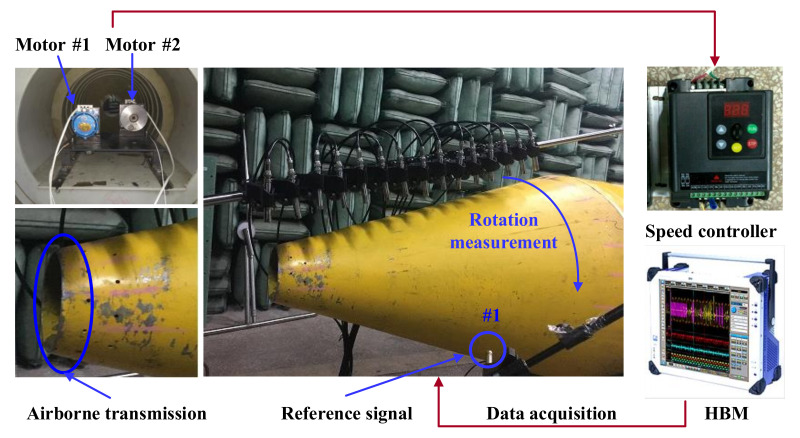
Measurement scheme.

**Figure 24 sensors-21-07150-f024:**
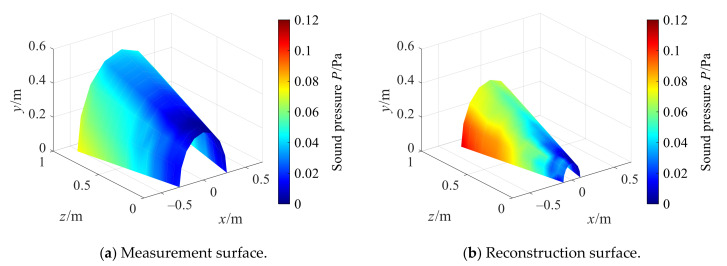
Actual measuring sound pressure.

**Figure 25 sensors-21-07150-f025:**
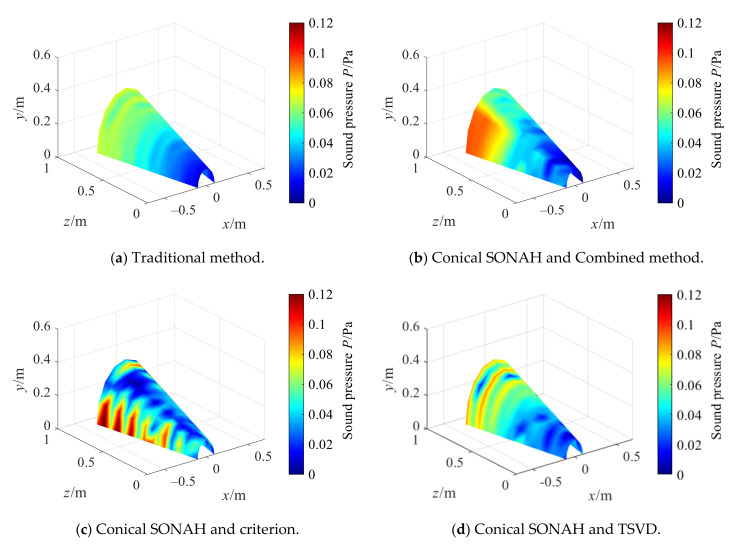
Reconstruction by conical SONAH.

**Figure 26 sensors-21-07150-f026:**
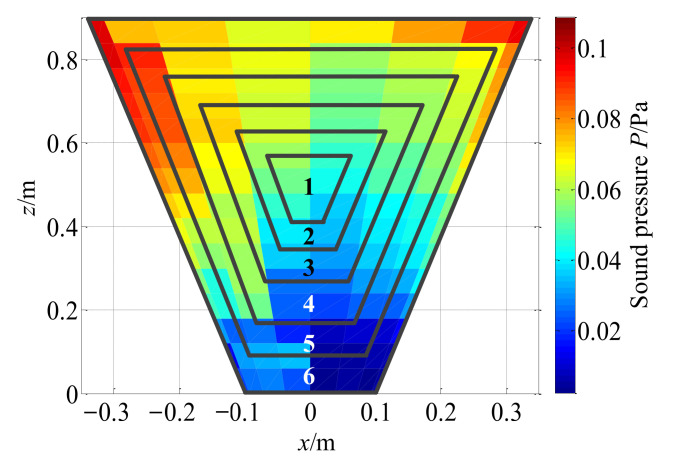
Different regions on the reconstruction surface.

**Figure 27 sensors-21-07150-f027:**
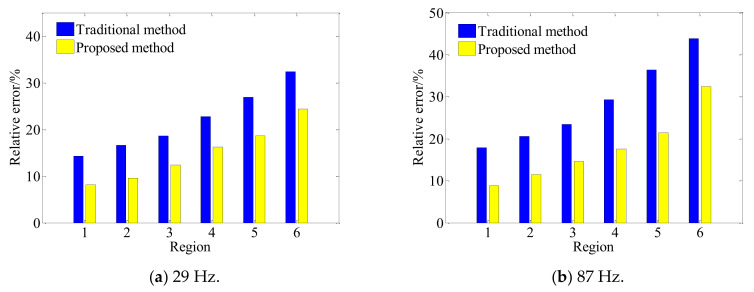
Reconstruction by the traditional and proposed method.

**Figure 28 sensors-21-07150-f028:**
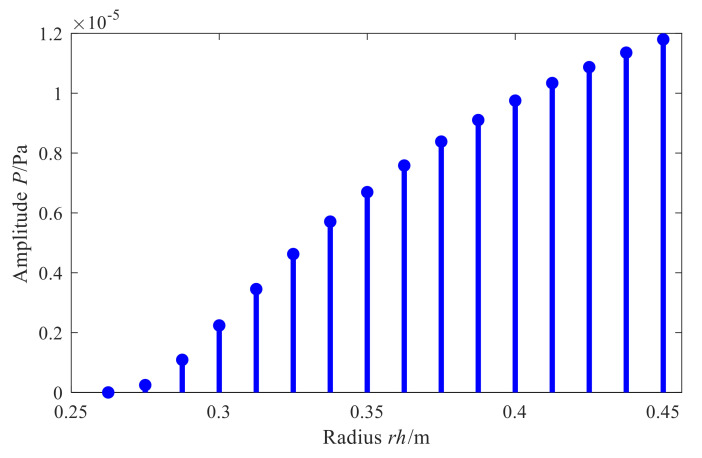
Amplitude variation of $\Phi_{Km}^{\beta }(r)$.

**Table 1 sensors-21-07150-t001:** The influence of singular value contribution rate on reconstruction error.

Contribution rate %	1	2	3	4	5	7	10	15	25
Relative error %	12.31	11.81	11.68	11.39	11.45	12.34	12.34	14.42	16.78

**Table 2 sensors-21-07150-t002:** Sensor parameters.

Name	Model	Frequency Range	Sensitivity	Range
Beijing SKC sound pressure sensor	MNP21	20~20 kHz	50 mV/Pa	18~136 dB

**Table 3 sensors-21-07150-t003:** Reconstruction error of different regularization methods.

Methods	Cylindrical SONAH	Conical SONAH and Hald	Conical SONAH and TSVD	Conical SONAH and Combined Method
Relative error %	34.70	24.68	18.32	11.14

## Data Availability

Not applicable.

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
