# Peer review of "Conical Statistical Optimal Near-Field Acoustic Holography with Combined Regularization"

_sensors, 2021, doi:10.3390/s21217150_

Round 1
Reviewer 1 Report
This manuscript proposes a conical SONAH method based on the cylindrical SOANH to realize the sound field reconstruction of large conical surfaces. The theory of the proposed method is well described, and the simulation and experimental results verified its feasibility. When solving the inverse problem, the manuscript claimed that a regularization method combing the TSVD and Tikhonov regularization was proposed. This regularization method can significantly improve the accuracy of the reconstructed results. However, the reviewer concerns the originality of this combined regularization method, since it has been presented in another paper (He et al., The application of regularization technique based on partial optimization in the nearfield acoustic holography, Acta Acustica, 2011: 36(4): 419-426). The use of this regularization method to the conical SONAH is acceptable, however, the original reference should be mentioned.
Some other comments:
- In the introduction section, it is stated that regularization is a method of overcoming the ill-posedness of inverse problem and some regularization methods are mentioned. However, it should be mentioned that the Morozov discrepancy principle, GCV and L-curve method are not the regularization methods but the methods for determining the regularization parameters.
- In the simulation section, too many figures were provided. It seems that some of them are unnecessary. For example, Figures 5-8, Figure 9 and Figure 10 describe almost the same thing. Figure 9 is enough.
- In the simulation and experimental sections, very few frequencies and SNRs are considered. Please provide the error curves versus frequency and SNR.
- Figure 21 provides the reconstruction without regularization. This result seems meaningless here, since it has been a common sense in the field of NAH.
- The experiment was carried out in a semi-anechoic chamber environment. I wonder if the reflection from the ground has been considered. If not, how to deal with this reflection in the manuscript.
- The section of 4.3.1 test data preprocessing can be deleted, since it can be found in any book on NAH.
- For a conical structure, the patch ESM is also a very effective method. What is the advantage of the proposed conical SONAH method?
- Some highly related references to SONAH and ESM are lost.
Reviewer 2 Report
The first part is very long and should be summarized and highlighted the contributions of the authors to the issue.
The paper is very long and often it is not clear what are the contributions of the authors to the issue.
Figures 5 - 8 explain how these graphs were obtained.
Figures 17 - 19 explain how the acoustic measurements were followed.
Figures 24 explain how the represented results were obtained.
Figures 27 and 28 lack the geometrical dimensions of the object (submarine!).
Where were the acoustic measurements performed? in an anechoic chamber? you should better explain how this environment is made.
You should explain how you performed the acoustic measurements the type of noise you measured, you can also represent acoustic measurement in the frequency domain.
Does the object you measure move in water?
Did you make the measurements in the air? Is there any difference?
You can describe these issues if the medium changes.
How the apparatus you used is made. How is the antenna with the microphones made? How did you calibrate the system?
Was the acquisition software developed by the authors?
The chapter where the acoustic measurements are described should be increased.
The number of references also increases.
Round 2
Reviewer 1 Report
The manuscript has been improved. The first author's name of Ref. 32 is incorrect. Please check it before publication.Author Response
Please see the attachment.
Reviewer 2 Report
accept
